# Mycotoxins of Concern in Children and Infant Cereal Food at European Level: Incidence and Bioaccessibility

**DOI:** 10.3390/toxins14070488

**Published:** 2022-07-15

**Authors:** Cheila Pereira, Sara C. Cunha, José O. Fernandes

**Affiliations:** LAQV-REQUIMTE, Laboratory of Bromotology and Hidrology, Faculty of Pharmacy, University of Porto, Jorge de Viterbo Ferreira 228, 4050-313 Porto, Portugal; cheila.vpereira@gmail.com (C.P.); josefer@ff.up.pt (J.O.F.)

**Keywords:** food toxins, mycotoxins, infancy/childhood nutrition, quality control, bioaccessibility, chromatography

## Abstract

Cereals are of utmost importance for the nutrition of infants and children, as they provide important nutrients for their growth and development and, in addition, they are easily digestible, being the best choice for the transition from breast milk/infant formula to solid foods. It is well known that children are more susceptible than adults to toxic food contaminants, such as mycotoxins, common contaminants in cereals. Many mycotoxins are already regulated and controlled according to strict quality control standards in Europe and around the world. There are, however, some mycotoxins about which the level of knowledge is lower: the so-called emerging mycotoxins, which are not yet regulated. The current review summarizes the recent information (since 2014) published in the scientific literature on the amounts of mycotoxins in infants’ and children’s cereal-based food in Europe, as well as their behaviour during digestion (bioaccessibility). Additionally, analytical methods used for mycotoxin determination and in vitro methods used to evaluate bioaccessibility are also reported. Some studies demonstrated the co-occurrence of regulated and emerging mycotoxins in cereal products used in children’s food, which highlights the need to adopt guidelines on the simultaneous presence of more than one mycotoxin. Although very little research has been done on the bioaccessibility of mycotoxins in these food products, very interesting results correlating the fiber and lipid contents of such products with a higher or lower bioaccessibility of mycotoxins were reported. LC-MS/MS is the method of choice for the detection and quantification of mycotoxins due to its high sensibility and accuracy. In vitro static digestion models are the preferred ones for bioaccessibility evaluation due to their simplicity and accuracy.

## 1. Introduction

Nutrition in the first years of life is essential to provide optimal growth and development and to establish preferences and eating patterns [1,2,3]. At 6 months of age, breastfeeding should be gradually replaced by nutritionally adequate and safe complementary food, starting with small portions of food that gradually increase with child development [3,4]. From 6 to 23 months of age, children’s nutritional needs require that they eat foods from at least four of the subsequent groups per day: grains, roots and tubers, legumes and nuts, dairy products, meat and fish, eggs, vitamin A-rich fruits and vegetables, and other fruits and vegetables. If their diet lacks this diversity of food, there is a risk of potential micronutrient deficiency [3].

Infant cereals are, usually, among the first foods introduced as complementary foods during weaning because they can complement breastfeeding, guaranteeing the nutritional requirements of the infant, providing a large number of proteins, vitamins (B1, B2, B3, B6, B9, E, and K), minerals (iron, zinc, magnesium, sodium, potassium, and phosphorus), and bioactive compounds, and allowing iron fortification. Also, cereals provide non-digestible carbohydrates that are responsible for the development of microbiota (increasing the *Bacteroides* population). Due to their mild taste and semi-solid texture, cereals allow the transition from milk and the consequent acceptance of solid foods [5]. There are several different types of cereal-based products available for children, and they are composed of different base cereals, such as wheat, maize, barley, rice, and oats, which can be mixed with chocolate, honey, fruits, or nuts for the enhanced flavour and attractiveness of the product.

Cereals used in infant food are usually subjected to meticulous quality control processes before their release to the market; however, crops are susceptible to toxigenic fungi during both pre- and post-harvest steps, which can produce small toxic chemicals: mycotoxins. The main fungal producers of mycotoxins belong to the genera *Aspergillus* spp., *Penicillium* spp., and *Fusarium* spp. Generally, the identification of the fungi responsible for mycotoxin contamination is difficult since a single fungal species may be able to produce different mycotoxins, while a certain mycotoxin can be produced by more than one species. For instance, mycotoxins such as aflatoxins (AFs), ochratoxins (OTA), citrinin (CIT) [6], and sterigmatocystin (STE) [7], the last two being not regulated in this type of matrix, are produced by the *Aspergillus* species, while trichothecenes (TCs) (group A: HT-2, and T-2 toxins, and group B: deoxynivalenol (DON)), zearalenone (ZEA), fumonisins B1 (FB1) and B2 (FB2), and the emerging mycotoxins beauvericin (BEA) [8] and enniatins (ENs) [7] are often produced by the *Fusarium* species. Ergot alkaloids are produced by *Claviceps* [9].

Mycotoxins occur along all cereal food chains and can have a myriad of acute and/or chronic detrimental health effects, such as immunosuppressive, carcinogenic, estrogenic, gastrointestinal, and kidney events [10,11,12]. Therefore, several countries have adopted regulations to limit mycotoxin exposure through food and reduce their levels to as low as possible. In Europe, EC (European Community) set different maximum limits for mycotoxin exposure for adults and for children (Table 1)—Commission Regulation (EC) No 1881/2006, Commission Regulation (EC) No 165/2010, and Commission Recommendation (2013/165/EU). While the “classical mycotoxins” are already well-known and regulated, there are other mycotoxins, i.e., the so-called emerging mycotoxins, that are not routinely determined nor regulated, even though there is evidence of their rapidly increasing incidence [13].

This manuscript aims to present for the first time a holistic review of mycotoxins in infants’s and children’s cereal-based foods found in Europe since 2014. To accomplish the objective, the amounts of mycotoxins, their bioaccessibility as well as the advantages and drawbacks of the analytical methods used for their determination will be provided. The impact of mycotoxins in infants and children and the protective effect of the ingestion of cereals against mycotoxicosis will be briefly summarized. Additionally, different factors that determine mycotoxins’ bioaccessibility will be highlighted in order to understand whether the fraction of mycotoxins released from the food matrix into the gastrointestinal tract upon digestion could induce toxic health effects in the children. We provide an overview of the gaps in research on mycotoxins in cereal-based foods for infants and children, to stimulate and improve future research avenues.

## 2. Materials and Methods

A review of the literature was performed by using the databases Scopus^®^ (www.scopus.com), Web of Science^®^, PubMed^®^, and Google Scholar. Several keywords were included to identify published works for mycotoxin incidence in cereal and cereal-based products for infants and children. These included “occurrence”, “mycotoxin”, “aflatoxin”, “ochratoxin”, “deoxynivalenol”, “fumonisin”, “patulin” “T-2”, “HT-2”, “enniatin”, “beauvericin”, “cereal” or “cereal-based products”, “infant” or “children”, and “Europe”. The terms were searched across document titles, types, abstracts, and keywords lists across all years since 2014. Thirteen studies met the selected criteria and were included in the revision. The articles on bioavailability of mycotoxins in this matrix were selected using keywords such as “mycotoxin”, “bioaccessibility” or “bioavailability”, “cereal” or “cereal-based products”, “infant” or “children”; since only four studies were found, no timeframe was established. Data from the studies were divided into different sections: major mycotoxins in cereals, bioaccessibility of mycotoxins in cereal-based infants’ and children’s food and strategies for its reduction, methods for mycotoxin analysis in food and bioaccessibility studies, and gaps in the research of mycotoxins in infants’/children’s cereal-based food matrices (Figure 1).

## 3. Mycotoxicosis in Infants and Children

Mycotoxin contamination is characterized by overtime exposure with the repeated consumption of the same kind of products. Children are more exposed to acute mycotoxicosis because they have stricter dietary patterns, consume some specific food products in larger amounts than adults (per kg of body weight), and their metabolism is not totally developed, so their system is more sensible [9,14,15,16]. However, as the levels of mycotoxins in food are usually low, the long-term effects are rarely seen in children [17,18].

Among children and infants, the most characterized symptoms are recurrent vomiting, caused by exposure to AFs, fumonisins, T-2, patulin (PAT), and DON, bone marrow failure, a well-recognized effect of contamination by TCs [15,19], in addition to recurrent apnoea, pneumonia and/or acute pulmonary haemorrhages, when the exposure route is inhalation [19]. Although unusual in children, there are some long-term consequences due to exposure to these toxins, such as hepatocellular carcinoma by AFs [10], oesophageal cancer and neural tube defects by fumonisins [10,11,12], renal cancer and Balkan nephropathy by OTA [10,20], estrogenic effects by ZEA [21,22], and ergotism by EAs [10] (Figure 2).

Damage to the gastrointestinal system may also only be revealed in the long term. Mycotoxins affect the digestive system in two ways: first by altering the gut microbiota due to a direct toxic effect on the microorganisms and second by altering the structures of the intestine, e.g., TCs and PAT can affect the intestinal barrier, causing the weakening of the permeability and integrity of the intestinal epithelium, resulting in inflammation [17,23,24].

## 4. Protective Effects of Cereals against Mycotoxin Exposure

Cereals, especially whole-grain, provide several compounds with health-protective mechanisms that may diminish the effects caused by mycotoxins’ toxicity. Particularly, wheat bran has shown consistent cancer-protective properties in human and animal experimental models [25]. The anti-carcinogenic properties of whole-grain foods are mainly due to antioxidant and anti-inflammatory compounds, such as phenolic acids [26,27,28], flavonoids, carotenoids [29], vitamin E [30], n-3 fatty acids [31], phytic acid [32,33], and selenium [34]. Whole-grain cereals are also an indirect source of short-chain fatty acids such as acetate, butyrate, and propionate, which have cancer-preventing properties by lowering intestinal pH and reducing the ability of bile acids to act as carcinogens [35,36]. Moreover, the presence of insoluble fibres in whole-grain cereals increases colon transit time and faecal bulking, which leads to the dilution of carcinogens and reduces their interaction with epithelial cells, resulting in cancer prevention [35]. Dietary fibres can also adsorb xenobiotics, resulting in diminished exposure and absorption by the digestive tract [37]. Each of these compounds acts complementary to each other, which enhances their protective action [38].

## 5. Major Mycotoxins in Cereals

There are several studies on the prevalence of mycotoxins in cereal-based products for young infants and children, as reported in some previous review manuscripts [39,40,41]. In Table 2, there is a summary of the data from those reviews with special focus on the incidence of mycotoxins in cereal and cereal-based foods for infants and children commercialized in Europe.

From 2014 to 2021, thirteen studies reported the presence of mycotoxins in cereal-based infants’/children’s food products in Europe. The majority of the studies verified that TCs, particularly DON, are the most prevalent mycotoxins [42,43,44,47,48,50,51,54]. In some of these studies, DON levels were above the maximum limit established by EC (200 µg/kg) [42,43,48,50]. HT-2 was detected in two studies [42,51], one of which found two samples above the established maximum limit (15 µg/kg) [42], and T-2 was also found in two studies at low levels in the first in four samples [51], and in the last in nine samples [53]. Nivalenol (NIV), a TCs unregulated mycotoxin, was found in the cereal samples in five studies, namely in four samples in the studies by Juan et al., 2014 [42], and Martins et al., 2018 [47], in three samples in the study by Postupolski et al., 2019 [51], and in two samples in the study by Braun et al., 2021 [53].

OTA was also a very prevalent mycotoxin in cereal-based food for young people, being reported in eight out of thirteen studies [42,44,45,46,47,48,51,52], usually in levels below the maximum limit established by the EC (0.5 µg/kg). In the study by Juan et al., 2014 [42], this mycotoxin was present in 15 of 75 (20%) samples of cereal-based baby food. Assunção and team [46] found that 40% of the samples presented co-occurrence of PAT and OTA, which underlines the necessity of establishing a maximum limit of PAT in processed cereal-based foods. Martins et al., 2018 [47] reported the presence of OTA in 18 samples. Assunção et al., 2018 [48] analysed different types of samples and reported the presence of OTA in 18 samples of breakfast cereals, 10 samples of infant flours, and 6 samples of biscuits. Lastly, Postupolski et al., 2019 [51], detected the presence of this mycotoxin in four samples.

Regarding AFs, despite being one of the most studied mycotoxins, only four studies reported their presence in cereal-based food samples marketed for children [44,47,48,50]. AFB1 was above the maximum limit established by the EC of 0.1 µg/kg in the works of Martins et al., 2018 [47] and Herrera et al., 2019 [50], both in six samples. In the last study, AFB2 (*n* = 1), AFG1 (*n* = 6), and AFG2 (*n* = 1) were also found.

Fumonisins and ZEA were found less in cereal-based baby foods, and when detected, the levels were below the maximum limits indicated by the EC, 200 µg/kg and 20 µg/kg, respectively [44,47,48,51]. In the study by Postupolski et al., 2019 [51], only one sample presented positive results for fumonisins, and 14 samples were contaminated with ZEA, while in the studies by Assunção et al., 2018 [48] and Martins et al., 2018 [47], both fumonisins and ZEA had a similar incidence of 19 and 15 positive samples, respectively.

ENs were quantified in cereal-based samples marketed for children in two works (Juan et al., 2014 and Braun et al., 2021), but their incidence was lower when compared to other regulated mycotoxins. ENB was the most detected enniatin, mostly in wheat cereal samples (*n* = 7) in the study by Juan et al., 2014 [42] and in 21 samples in the study by Braun et al., 2021 [53]. Even so, co-occurrence with these emerging mycotoxins demonstrates the importance of the establishment of new guidelines and the necessity of more studies in these matrices.

Other emerging mycotoxins such as STG [53], Alternaria toxins (AOH, AME, TEN, TA, ATX I or ATPL) [49,53], T2-tetrol [43], and AFL [53], were also found in the studies reported here, however, with a much lower frequency.

## 6. Bioaccessibility of Mycotoxins in Cereal-Based Infants’ and Children’s Food and Strategies for Its Reduction

Despite the knowledge already acquired about the presence of mycotoxins in food products, the amounts that are effectively absorbed by the human body are not known, as little is known about their bioavailability and bioaccessibility once ingested [55,56,57] (Figure 3).

Mycotoxin bioavailability is the measurement of the amount of mycotoxin that can reach circulation after absorption by the intestinal epithelium (Figure 2). Different food matrices can affect the bioaccessibility of a mycotoxin; however, the metabolism and absorption depend on the mycotoxin itself [55,57,58].

There are few studies on the bioaccessibility of mycotoxins in children (Table 3), but it is known that a child may be more susceptible to significant damage to the intestinal enterocytes caused by the presence of these toxins in the intestinal fluid when compared to adults [15,58,59].

Kabak and colleagues evaluated the effects of probiotic bacteria on bioaccessibility of AFB1 and OTA alone and together in infant formula, with an in vitro static digestion model. The bioaccessibility levels of AFB1 and OTA differ from each other, 88–94% for AFB1 and 29–32% for OTA, which shows that the bioaccessibility depends on the mycotoxin [60]. The authors found that the effect on bioaccessibility depended on bacteria species and type of mycotoxin, with *Lactobacillus acidophilus* and *L. casei* Shirota being more effective in reducing the bioaccessibility of AFB1, while no effect was observed on the bioaccessibility of OTA.

In 2012, Raiola’s team evaluated the bioaccessibility of DON in six Italian commercial pastas marketed for young children using an in vitro static digestion model that simulated child physiology. The percentage of DON in the gastric fluid ranged from 2.12 ± 0.11–38.41 ± 2.95%, and the bioaccessibility of DON after the duodenal process ranged from 1.11 ± 0.01–17.91 ± 0.80%. The authors concluded that the bioaccessibility of DON is high enough to produce toxicity and cause higher damage to children, especially due to the small dimension of the intestinal epithelium [59].

Prosperini and colleagues [61] used an in vitro static digestion model to study the bioaccessibility of ENA, ENA1, ENB, and ENB1 in 14 samples of breakfast cereals, cookies, and bread. The samples were spiked with the mycotoxins at 1.5 and 3.0 µmol/g. Their results showed no significant differences between spiked levels in the same type of samples for all mycotoxins except ENB1, where there was higher bioaccessibility in the samples spiked with 1.5 µmol/g (67.3 ± 2.7%). Among the mycotoxins, ENA has the highest bioaccessibility, followed by ENB1, ENB, and ENA1. There are, however, differences in bioaccessibility between similar types of samples that may be related to the chemical structure of the mycotoxins, the food product composition, or matrix. When comparing different cereals, the ones with higher fibre content showed lower bioaccessibility, as the fibres combined with the mycotoxins, leaving less available to produce toxic effects. The inclusion of dietary fibres offers some protection against mycotoxicosis, resulting in a cheap and effective method for the detoxification of feed and food products. Likewise, when comparing different samples with different lipid content, the authors noticed that food products with a higher lipid content show lower bioaccessibility. When the mycotoxins interact with the fat content of the food product, they are not released completely.

The most recent study on bioaccessibility of mycotoxins in cereal-based food for children is from Assunção et al. [46], who studied the bioaccessibility of PAT and OTA in processed cereal-based foods with an in vitro static digestion model. The authors analysed six samples (three with fruit and three without), which were spiked with 20 µg/kg of PAT and 1 µg/kg of OTA, and they verified that the bioaccessibility values of OTA were significantly higher than those of PAT, 106 ± 0.6%, and 56 ± 2.7%, respectively. When PAT was assayed alone or combined with OTA, there was no statistical difference in the values of bioaccessibility, with mean values of 52 ± 4.2% and 56 ± 2.7%, respectively. There was, however, a statistical difference when assaying OTA alone or combined with PAT, with mean values of 100 ± 1.1% and 106 ± 0.6%, which means that OTA demonstrated higher bioaccessibility when in a mixture with PAT; the values above 100% may be due to possible interactions between food matrix, mycotoxins, and digestive fluids.

In vitro biotransformation studies with cell lines offer deeper insight into the effect of the digestion process on mycotoxins and their bioavailability, and consequently, on the impact of these contaminants on the intestinal cells. The biotransformation process of mycotoxins is composed of two phases: in the first, mycotoxins may suffer oxidation, reduction, or hydrolyzation by cytochrome P (CYP) enzymes present in the endoplasmic reticulum and mitochondria, and in the second phase, the resulting metabolites may form conjugates with glutathione, glucuronic acid, and sulphate [62]. The Caco-2 (colorectal adenocarcinoma cell) cell line is the preferred in vitro model method to analyse the intestinal absorption of compounds because these cells form a continuous monolayer with tight junctions that mimic the intestinal wall [63]; also, the ability to express various phase I and II enzymes makes it a good option for biotransformation studies. Other models, such as IPEC-J2 (intestinal porcine epithelial cell), are also used [64,65]. Epithelial integrity and the organization of the tight junction in the cell monolayer are evaluated by monitoring the trans-epithelial electrical resistance (TEER) and the expression of tight junction proteins.

The biotransformation processes of the most studied mycotoxins are already known. AFB1, while not reactive by itself, after bioactivation by different CYP enzymes, forms different metabolites with different degrees of toxicity, such as AFB1-8,9-exposide (AFBO) [66], which is very toxic, with mutagenic and carcinogenic characteristics via the action of CYP450, aflatoxicol (AFL), which is mildly toxic via a ketoreduction, and AFM1, which is also mildly toxic via hydroxylation. The detoxification process of AFBO and AFM1 involves a phase II reaction, conjugation with glutathione-by-glutathione S-transferase (GST) [67,68]. Studies reported that AFB1 reduces TEER in a time-dependent manner in Caco-2 cell line culture [69,70,71] and significantly decreases mRNA expression of tight junction proteins, which are essential for maintenance of adhesive and selective permeability characteristics of the intestinal epithelium [70,71], while AFM1, which is less toxic than AFB1, damaged epithelial integrity to a lesser extent [71,72]. The metabolites resulting from OTA biotransformation are usually less toxic than the original compound [73]. The most common metabolites formed are OTα, formed by the action of carboxypeptidases, 4-hydroxy-ochratoxin A (4-OH-OTA), and 10-hydroxyochratoxin A (10-OH-OTA), formed by reaction with CYP450 enzymes [66,74,75]. A study by Huang et al., [72] showed that OTA has the capacity to reduce TEER in a dose-dependent manner, both individually and in conjugation with AFM1. A study by Alizadeh et al., 2019 [76] demonstrates that OTA reduced TEER in a dose- and time-dependent manner and also decreased the expression of tight junction proteins. The same effect was demonstrated in a different type of cellular model, porcine intestinal cell line IPEC-J2 [77]. DON metabolites are formed by phase II metabolism with glucuronide and sulphate conjugation. The most common metabolites are 3-acetyl-DON, 15-acetyl-DON, diepoxy-deoxynivalenol (DOM-1), DON-3-guccoside (D3G), DON-sulfonates and DON-3-glucuronides [78,79,80]. In studies with Caco-2 cell lines, there was no metabolization of DON, and TEER reduction was apparent only with high exposure times [81,82]. Kadota et al., 2013 compared DON and two metabolites, 3AcDON and 15AcDON, on their intestinal transport in Caco-2 cell line. The authors found that all compounds reduced TEER; however, 15AcDON induced a significantly higher loss of epithelial integrity than DON or 3AcDON [83]. In IPEC-J2 cell line, DON was capable of significantly reducing tight junction protein expression, reducing stability, and increasing the degradation of existing tight junction proteins, leading to increased epithelial permeability [65]. The effect of DON on proliferating and differentiated Caco-2 cell lines was evaluated by Luo et al., 2021. This mycotoxin induces a reduction of TEER in differentiated cells and delays the establishment of TEER in proliferating cells, which may lead to a reduction in the renewal and repair rates of the intestinal epithelium [84]. T-2 toxin can be metabolized by hydroxylation, hydrolysis, and deepoxidation reactions (phase I) and by conjugation (phase II). The most common metabolites formed are HT-2, neosolaniol (NEO), and T-2 triol. While in phase I, CYP450 enzymes are the major contributors, and in phase II, the conjugation is most common with glucuronide [79,85]. T-2 induces a reduction in TEER values in cytotoxic concentration in IPEC-J2 cells, while non-cytotoxic concentrations have little effect [64]. The study by Romero et al., 2016 demonstrated that T-2 was capable of decreasing TEER and the expression of tight junction proteins in a concentration-dependent manner [70]. FB1 is not metabolized by the action of CYP450 enzymes and can act as an inhibitor of specific CYP450 enzymes, such as CYP2C11 and CYP1A2. When FB1 is acetylated, the resulting N-acetylated metabolites are more toxic than the original FB1 [62,66,86,87]. This mycotoxin also affects epithelial integrity by downregulating tight junction protein expression and decreasing TEER. In a study evaluating four mycotoxins (AFB1, FB1, T-2, and OTA), FB1 caused the second-highest reduction of TEER, only surpassed by AFB1 [70]. In the case of ZEA, the most common metabolites are α-zearalenol (α-ZEA), which has a higher affinity for oestrogen receptors in humans, β-zearalenol (β-ZEA) via action of CYP450 enzymes, and zearalenone-14-glucoside (ZEA14Glc) via glycosylation [88,89]. The study by Videmann et al., 2008 demonstrated the metabolization of ZEA into mainly α-ZEA, followed by β-ZEA and ZEA glucuronide, leading to a higher susceptibility to the effects of this mycotoxin [90].

Besides good agricultural practices on crop cereals production, different types of food processing such as washing, roasting, grinding, cooking, radiation, and others [10,91] are strategies used to reduce the occurrence of mycotoxins in cereal food staples. For instance, Meca and co-workers studied the influence of heat treatment on the degradation of BEA, at 160, 180, and 200 °C and at different incubation times, 3, 6, 10, 15, and 20, in crispy breads with different flours. They found a 43% reduction of BEA bioaccessibility at 160 °C, 69% reduction at 180 °C, and 87% reduction at 200 °C, with increasing reduction with longer time of incubation [92]. There are, also, strategies to reduce the bioaccessibility of mycotoxins during digestion, such as the use of probiotic bacteria [58], chemical reduction, and food processing [93]. In 2009, Kabak and colleagues reported a reduction of 37% and 73% for AFB1 and OTA, respectively, in the presence of Lactobacillus and Bifidobacterium bacteria in pistachios, buckwheat, and infant food [60]. Another study on loaf bread samples showed a reduction of AFB1 (15–98%) and AFB2 (11–98%) bioaccessibility in gastric and duodenal digestion, with the presence of probiotic bacteria [94]. Isothiocyanates are compounds present in plants from the Brassicaceae family that have strong antimicrobial properties. Luciano and co-workers studied the reduction of BEA in cereals with BITC (benzyl isothiocyanate) and PITC (phenyl isothiocyanate) fumigation and found a significant reduction, with the highest rate at 92.5% in rice kernels [95].

## 7. Methods for Mycotoxin Analysis in Food and Bioaccessibility Studies

### 7.1. Analysis of Mycotoxin in Food

Identification and quantification of mycotoxins in food are critical steps to support production quality control and health risk assessment. Selective and robust methods are needed for mycotoxin analysis because they are very complex molecules with vastly different chemical structures that are present in a range of matrices, usually in low concentrations [96]. Co-occurrence of different mycotoxins is also an issue to consider when analyzing a product.

Mycotoxin analytical methods include several steps such as sampling, homogenization, extraction followed by a clean-up step to decrease and/or eliminate matrix compounds, sample concentration, separation, and detection. The detection step is performed either via a chromatographic technique with different detectors or via an immuno-chemical method [97,98]. More details about the sampling, sample preparation and detection method see Appendix A.

### 7.2. Bioavailability and Bioaccessibility Analysis

Bioaccessibility assessment can be carried out by in vivo or in vitro assays. In vivo approaches are very complicated due to analytical and ethical questions, are very time-consuming and require specific planning, resources, and a high level of experimental control. Although less realistic, in vitro approaches have a wider experimental scope and are simpler to perform while still producing useful results [57].

In vivo studies can be divided in two groups: overall balance studies and determination of substances (or metabolites) in relevant tissues, while in vitro studies, can be divided into static and dynamic processes [57]. Static models mimic the digestive transit by the sequential exposure of the food product to the conditions of the stomach and small intestine, whereas dynamic models simulate the gradual transit of the food product through the different processes of the digestive tract, ensuring a more realistic simulation of the digestive process. These last models simulate the gastric emptying patterns, food transit times, pH value modifications, different concentrations of enzymes, bile salts and electrolytes, water absorption percentage, and in some models, the microbial activity in the different compartments of the gastrointestinal tract [58]. Some models simulate the digestive process in the stomach and small intestine of monogastric animals while other models mimic the large intestine and include microbiota derived from the intestine. As mycotoxin absorption occurs in the small intestine, large intestine models are not used in these studies [57,58].

In vitro digestion models have been developed for the study of several different contaminants, based on human and animal physiology. Some requirements need to be considered to achieve a good methodology, such as the following: (1) representative of the physiological processes of the organism in the study; (2) biochemical reactions, hydrodynamics, and mechanical forces used must match with digestion kinetics; (3) simulation of fasted and fed conditions including proper an-aerobic conditions for the presence of gastrointestinal microorganisms; and (4) easy, robust, reproducible, and applicable methodology [55,57].

The main physiological components of in vitro digestion models are (1) saliva, because the digestion process starts in the mouth with a mechanical action of chewing aided by salivary digestion. This fluid is an exocrine secretion consisting of 99% water, electrolytes, and proteins. In most models simulated saliva is a simplified version of this fluid containing electrolytes, urea, and α-amylase. (2) Gastric juices, which are secreted by the gastric glands in the stomach, are composed by mucus, enzymes, and aqueous component, and in models, they are simulated by a strong decrease of pH and addition of pepsin and electrolytes. (3) Intestinal juices, because after gastric digestion, the food is released into the duodenum, where enzymes from the pancreas and bile from the liver continue the digestion process, and the simulated fluid is composed of electrolytes, pancreatin, and bile salts. Other important components in the digestion models are temperature, peristalsis, incubation time, and microbial interactions, the last not usual component in static models for mycotoxins because these compounds are absorbed in the small intestine [55].

All studies on bioaccessibility reported here (Table 4) used the in vitro static digestive model, with three steps (mouth, stomach, and small intestine digestion) because it is an easier and cheaper model. Kabak et al., 2009 [60] used a method developed by Versantvoort et al., 2005 [99]. The process starts with adding simulated saliva, pH 6.8, to a food sample and incubation for 5 min at 37 °C. Then, artificial gastric juice, pH 1.3, was added, following 2 h incubation at 37 °C. Finally, artificial duodenal fluid at pH 8.1 is added and the mixture is incubated for another 2 h at 37 °C. After digestion, the mycotoxin levels were evaluated in the extract. Raiola et al., 2012 [59] and Prosperini et al., 2013 [61] used a similar method adapted from Gil-Izquierdo et al., 2002 [100], with small alterations to components of the artificial fluids and the pH in each phase of digestion. The first step combines oral and gastric digestion, where the pH starts at 6.5 and is then adjusted to 2, and in the duodenal digestion, the pH is 6.5. After each digestion, the extract is obtained for mycotoxin extraction and detection. For the child digestion model, in the study of Raiola and team [59], there are slight modifications, where the pH of the stomach is 3 and the amount of pepsin, pancreatin, and bile salts were reduced. Assunção and team [46] used a method adapted from Minekus et al., 2014 [101], a standardized method, very similar to the one used by Kabak’s team, with a pH of 7 in the oral digestion, 3 in the gastric digestion, 7 in the intestinal digestion, and some alterations to the components of the artificial digestion fluids, as reported in Table 4. After the intestinal digestion, the mycotoxin levels were quantified in the extract.

## 8. Gaps in the Research of Mycotoxins in Infant/Children Cereal-Based Food Matrices

This review of the available research on mycotoxin quantification and bioaccessibility in cereal-based children and infant food products makes it clear that this is a topic that requires much more attention.

The majority of the studies on mycotoxin determination and quantification in these types of food products reported the presence of regulated mycotoxins, in many cases presenting values above the EU legal limits, and the co-occurrence of several compounds. Although only three studies have evaluated the presence of emerging non-regulated mycotoxins, positive results were always found, which leads to the possibility of a higher mycotoxin presence in cereal-based food products than the one reported in the studies based on the assessment of only regulated mycotoxins.

The type of samples used was quite diversified over the years, with almost all studies differentiating the samples by cereal composition, such as wheat, rice, barley, or maize, or by the presence of fruits and/or cocoa and presenting results in relation to that differentiation. However, only two studies (Herrera and Braun) mentioned products with cereals from organic/biologic cultures [50,53], and only Herrera et al., 2019 [50] presented the results of these specific products. There is an ever-increasing search for organic and biological products for their assumed health benefits, and more families are starting to feed their infants and children with homemade products and formulas, leading to the necessity of more research on these types of products. Braun et al., 2021 [53], diversified their sample pool by also quantifying mycotoxins in raw flours samples intended for production of infant foods. Some studies present other types of cereal-based products for children such as sweet cakes [52] and biscuits [48], highlighting the necessity to add other cereal-based products for children such as cereal bars and cookies. It is also possible to see some distinctions between cereal-based food products for babies, for infants, and for children, as with different age groups there are different nutritional needs, different eating habits, and different susceptibilities. Some studies [48,50,54] clearly present their results with a separation of these types of products, where the first team divided their samples into infant cereals and breakfast cereals [48], the second team considered two types of samples, gluten-free samples for babies from 4 to 6 months of age and multi-cereals for infants aged 7 to 12 months [50], and the last team separated their samples by brand, type of cereal, and consumption age [54].

Bioaccessibility studies on cereal-based infant food are scarce (only four in total as far as we know) and cover a small number of mycotoxins. For instance, ZEA, T-2, HT-2, AME, and TA, which were found in several studies [42,44,47,48,49,53], were never accessed for their bioaccessibility. Despite these types of studies being time-consuming, they do provide valuable information on the potentially harmful effect of mycotoxins on infants and children. In the study by Prosperini and colleagues [61], they reported that in the same type of food, a different composition can change the bioaccessibility of a mycotoxin; for example, a higher percentage of fibre or fat can result in lower bioaccessibility. This highlights the importance of carefully choosing the type of matrix when analysing the effect of a mycotoxin.

The co-occurrence of different mycotoxins [42,46,47,50,53] and whether they act as antagonists or synergists of each other needs to be further researched. More bioavailability studies, with more and different matrices, and other mycotoxins, regulated and non-regulated, will be of extreme importance for the evaluation of the cumulative effect of the compounds in the organism. It is also important to note the lack of studies that correlate the health benefits of cereal ingestion and their protective action against mycotoxins.

## 9. Final Considerations

Child nutrition is of extreme importance to guarantee the nutritional and energy requirements for proper growth and development. Cereals are one of the best types of food for the weening period in children, as they offer the most important nutrients, are easy to digest, and have a tolerable texture to allow an easier transition to solid foods.

Mycotoxin occurrence in food and processed food products is a worldwide issue due to its high prevalence, particularly in cereals. Nowadays, there are several EU regulations on maximum limits allowed for the most recognized mycotoxins, such as AFs, OTA, ZEA, FMs, and PAT. These regulations are especially severe regarding the limits required for food and food products destined for children, as they are more susceptible to the effects of mycotoxin contamination.

This review about the presence of mycotoxins in cereal-based infant foods commercialized in Europe shows that besides the regular presence of “classical” mycotoxins, which in some cases are above the legal limits, the co-occurrence of the so-called emerging mycotoxins that are not yet regulated is quite common. This finding underlines the necessity of a careful re-evaluation of current guidelines, as these only take into consideration the individual effect of each mycotoxin. Moreover, regulated mycotoxins are the only ones regularly tested for quality control purposes, which may lead to a misguided belief that infant food products are within the safety limits.

Few studies were reported on the bioavailability and bioaccessibility of mycotoxins from cereal-based food matrices, which leads to a diminished understanding of the full effects of the presence of these compounds in infant foods. More studies are needed in this area to gather a consensus on the quality parameters required for the products being marketed for and consumed by children in Europe. The favored method for bioaccessibility analysis is the static in vitro digestion model, due to its cost-effectiveness ratio. It is predictable that in the future, there may be wider use of dynamic digestion models because they offer a more realistic approach to the digestion process.

## Figures and Tables

**Figure 1 toxins-14-00488-f001:**
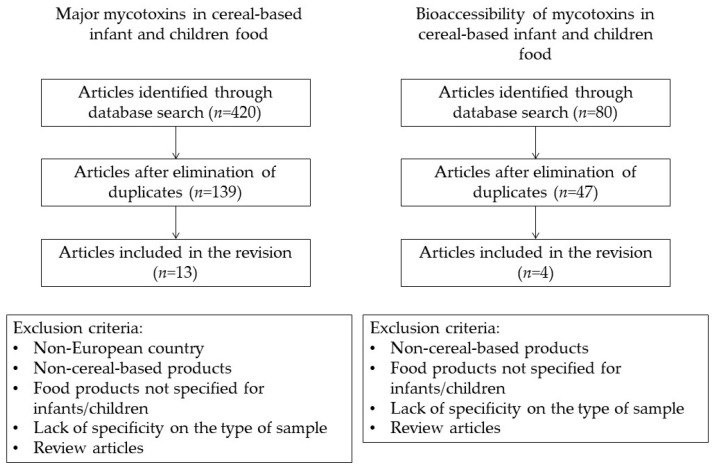
Methodology description diagram.

**Figure 2 toxins-14-00488-f002:**
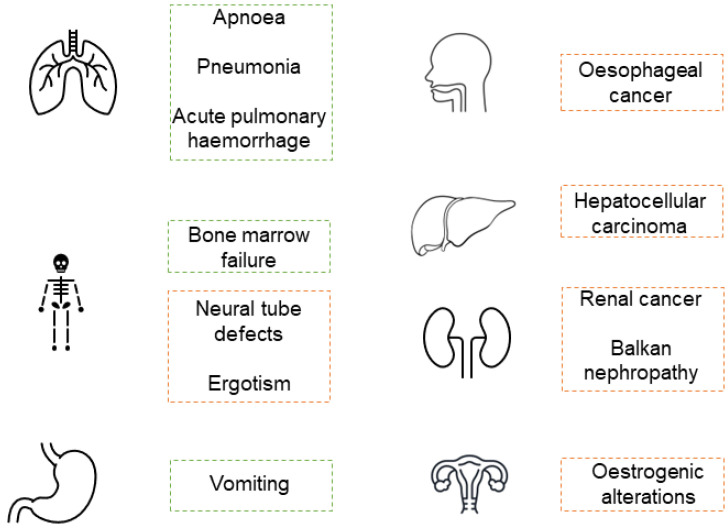
Most common mycotoxicosis health effects in children. Acute effects in green and chronic effects in orange.

**Figure 3 toxins-14-00488-f003:**
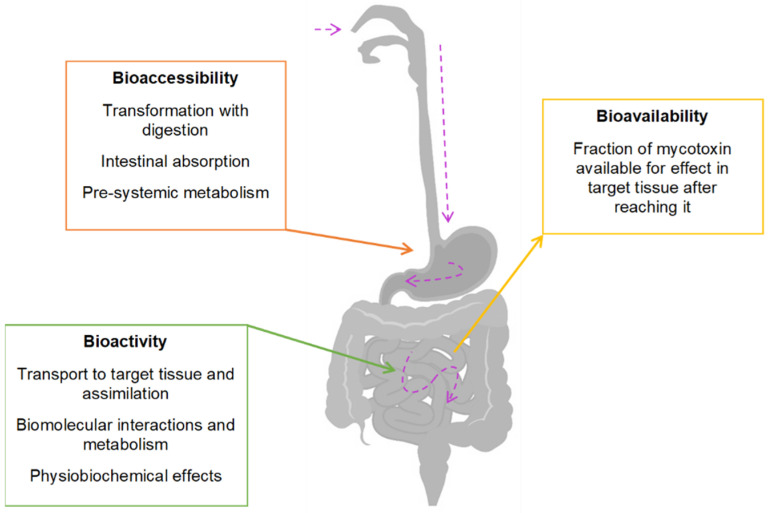
Bioaccessibility, bioactivity, and bioavailability definitions. Purple arrow—mycotoxin path after ingestion.

**Table 1 toxins-14-00488-t001:** Maximum levels of mycotoxins in cereals and cereal-derived products according to the European Commission.

Mycotoxins	Processed Cereal-Based Foods and Baby Foods for Infants and Young Children (µg/kg)
Aflatoxin B1	0.1
Ochratoxin A	0.5
Patulin *	10
Deoxynivalenol	200
Sum T-2 and HT-2 toxin	15
Zearalenone	20
Sum Fumonisin B1 and Fumonisin B2	200

* Baby foods other than processed cereal-based foods for infants and children.

**Table 2 toxins-14-00488-t002:** Occurrence of mycotoxins in cereal-based infant and young children food in Europe (2014 –2021).

Country	Sample	Mycotoxin	Total Samples	Positive Samples			References
				%	Mean (µg/kg)	Range (µg/kg)	
Italy	Infant formulas and baby food	OTA	75	20	0.06	0.050–0.120	Juan et al., 2014 [42]
		DON		25.3	102.60	1–268	
		NIV		4	19.91	5.5–235	
		FUS-X		24	146.51	5.5–604	
		HT-2		2.7	12.65	2–151	
		β-ZOL		6.7	2.5	2–23.2	
		ENB		13.3	101.30	5–832	
		ENB1		1.3	7.80	5–117	
		ENB4		5.3	38.08	5–311	
		ENA1		4	6.58	5–125	
		BEA		1.3	1.18	5–21.3	
Portugal	Cereal baby food (maize, wheat, rice, barley, rye, oat, sorghum, millet, spelt)	DON	9	44	173.13	0.37–270.57	Pereira et al., 2015 [43]
		15AcDON		11	30.94	2.50–30.94	
		T2-Tetrol		11	112.18	10.48–112.18	
		NEO		11	87.21	1.28–87.21	
Portugal	Breakfast cereals for children (maize, wheat, rice, and multi-grain)	AFB1	26		0.028	0.040–0.400	Assunção et al., 2015 [44]
		AFB2			0.002	0.030–0.300	
		AFG1			0.006	0.045–0.450	
		AFM1			0.012	0.100–1.000	
		OTA			0.026	0.200–2.000	
		DON			59	15–360	
		NIV			6	25–360	
		FB1			13	2.5–8.0	
		FB2			3	2.5–8.0	
Turkey	Baby food (cereal based supplementary foods for infants and children)	OTA	50	68	0.034–0.374	0.042–0.380	Hampikyan et al., 2015 [45]
Portugal	Children cereal-based food	PAT	20	75 50 40	2.33 0.061	3.2–40.0 0.2–2.0	Assunção et al., 2016 [46]
Portugal	Breakfast cereals	AFB1	26	69	0.013	0.003–0.130	Martins et al., 2018 [47]
		AFB2		27	0.004	0.001–0.011	
		AFG1		4	0.013	0.006–0.014	
		AFM1		12	0.017	0.011–0.240	
		OTA		69	0.040	0.006–0.100	
		FB1		58	12.5	0.06–67.0	
		FB2		38	4.2	0.12–14.0	
		DON		62	91.5	0.4–207.8	
		NIV		4	27.1	5.6–27.1	
		ZEA		19	0.7	0.12–5.6	
Portugal	Cereal-based children food	AFB1	26 breakfast cereals	73	0.036	NM	Assunção et al., 2018 [48]
		AFB2		46	0.07		
		AFG1		4	NA		
		AFM1		12	0.017		
		AFs		73	-		
		OTA		69	0.047		
		FB1		58	22.00		
		FB2		39	5.10		
		FMs		58	-		
		ZEA		73	1.20		
		DON		62	95.9		
		NIV		4	NA		
		AFB2	20 infant cereals (flours)	5	NA	NM	
		AFG1		10	0.014		
		AFM1		40	0.068		
		AFs		45	-		
		OTA		50	0.061		
		FB1		35	0.44		
		FMs		35	-		
		ZEA		30	0.48		
		DON		20	41.8		
		OTAOTA	6 biscuits	100	0.086	NM	
		DON		50	43.8		
Germany	Cereal-based baby food	AOH	19	36.8	0.89	4.73–7.13	Gotthardt et al., 2019 [49]
		AME		89.5	0.24	0.23–0.58	
		TEN		94.7	1	0.18–7.53	
		ATX I		15.8	0.17	NA	
		ATLP		5.3	0.24	NA	
		TA			50.2	5.66–221	
Spain	Cereal-based baby food	AFB1	60	11	0.03	0.02–0.23	Herrera et al., 2019 [50]
		AFB2		1	0.01	0.02–0.20	
		AFG1		6	0.02	0.02–0.16	
		AFG2		1	0.01	0.02–0.11	
		DON		12	37	33–245	
Poland	Cereal-based infant and children food	DON	302	17	>LOD ^a^ <LOQ ^b^	NM	Postupolski et al., 2019 [51]
		NIV		3			
		ZEA		14			
		OTA		4			
		HT-2		0			
		T-2		1			
		FB1		3			
		FB2		4			
Italy	Breakfast cereals Sweet cakes	OTA	84	2.38	1	NM	Capei et al., 2019 [52]
				35.7	1.34		
Austria and Czech Republic	Processed cereal-based infant foods	AFL	35	6	-	<LOQ–1.1	Braun et al., 2021 [53] *
		AFB1		-		-	
		STG		23	-	<LOQ–0.5	
		ZEA		3	0.24	1.2	
		DON		6	-	25–62	
		NIV		6	43	<LOQ–20	
		T-2		26	-	0.8–3.0	
		BEA		14	1.5	<LOQ–3.1	
		ENA		3	−1.9	<LOQ	
		ENB		11	0.7	<LOQ–2.1	
		ENA1		60	5.9	<LOQ–40	
		ENB1		26	3.9	<LOQ–10	
		FB1		20	4.8	<LOQ–8.3	
		AME		20	0.6	<LOQ–1.1	
		TA		31	48	<LOQ–124	
		TEN		34	0.9	<LOQ–1.5	
		ATPL		23	11	<LOQ–20	
Poland	Cereal-based baby foods	DON	110	9.09	107.8	62–148	Mruczyk et al., 2021 [54]

AFB1 (Aflatoxin B1), AFB2 (Aflatoxin B2), AFG1 (Aflatoxin G1), AFG2 (Aflatoxin G2), AFM1 (Aflatoxin M1), OTA (Ochratoxin A), DON (Deoxynivalenol), 15acDON (15-acetyldeoxynivelanol), NIV (Nivalenol), FUS-X (Fusarenon-x),T-2 (Mycotoxin T-2), HT-2 (Mycotoxin HT-2), T2-Tetrol (Mycotoxin T2-tetrol), β-ZOL (β-zearalenol), FB1 (Fumonisin B1), FB2 (Fumonisin B2), PAT (Patulin), ZEA (Zearalenone) ENB (Enniatin B), ENB1 (Enniatin B1), ENB2 (Enniatin B2), ENB4 (Enniatin B4), ENA (Enniatin A), ENA1 (Enniatin A1), ENA2 (Enniatin A2), BEA (Beauvericin), STG (Sterigmatocystin), NEO (Neosolaniol), AOH (Alternariol), AME (Alternariol monomethyl ether), TEN (Tentoxin), ATX I (Altertoxin 1), ATLP (Alterperylenol), TA (Tenuazonic acid) and AFL (Aflatoxicol). Maximum Limit (EC): AFB1—0.1 µg/kg; OTA—0.5 µg/kg; PAT—10.0 µg/kg; DON—200 µg/kg; ZEA—20 µg/kg; FB1 + FB2—200 µg/kg; T-2 + HT-2—15 µg/kg. Bold—above the maximum limit (EC). NA—not applicable; NM—not mentioned.^a^ DON—2.0, NIV—18.6, ZEA—6.1, OTA—0.07, HT-2—1.1, T-2—0.1, FB1—1.4 and FB2—0.5 µg/kg. ^b^ DON—6.5, NIV—61.9, ZEA—20.5, OTA—0.24, HT-2—3.7, T-2—0.3, FB1—0.4 and FB2—1.5 µg/kg. * AFL—0.25; STG—0.1; NIV—16; BEA—0.4; ENA—0.4; ENA1—0.4; ENB—0.4; ENB1—0.1; FB1—7.0; AME—0.6; TA—24, TEN—1.0 and ATPL—10 µg/kg [53].

**Table 3 toxins-14-00488-t003:** Bioaccessibility of mycotoxins in cereal-based infants’ and young children’s food.

Country	Sample	Mycotoxin	Total Samples	Bioaccessibility (%)	References
The Netherlands	Infant formula (spaghetti Bolognese) supplemented with 2 mL sunflower oil per 100 g of food	AFB1 OTA	2	88 ± 16–94 ± 8 29 ± 6–32 ± 4	Kabak et al., 2009 [60]
Italy	Commercial pasta	DON	6	2.12–41.5	Raiola et al., 2012 [59]
Spain	Breakfast cereals Cookies Breads	ENA ENA1 ENB ENB1	14	50 ± 3–80 ± 3 40 ± 2–64 ± 2 43 ± 3–70 ± 3 46 ± 3–74 ± 2	Prosperini et al., 2013 [61]
Portugal	Cereal-based food	PAT OTA PAT+OTA PAT+OTA	6	30 ± 3–77 ± 2 95 ± 0–105 ± 2 33 ± 1–64 ± 2 (PAT) 103 ± 1–109 ± 0 (OTA)	Assunção et al., 2016 [46]

**Table 4 toxins-14-00488-t004:** Methods for analysis of mycotoxins bioaccessibility in cereal-based food for infants and children.

Matrix	Mycotoxin	Digestion Model	Extraction	Detection Method	References
Infant formula	AFB1 OTA	Static in vitro digestion model: Oral phase (KCl/KSCN/NaH_2_PO_4_/NaSO_4_/NaCl/NaHCO_3_/urea/a-amylase/uric acid/mucin) Gastric phase (NaCl/NaH_2_PO4/KCl/CaCl_2_/NH_4_Cl/HCl/glucose/glucuronic acid/urea/glucosamine hydrochloride/BSA/pepsin/mucin) Intestinal phase (NaCl/NaHCO_3_/KCl/HCl/urea/CaCl_2_(2H_2_O)/BSA/bile)	Phosphoric acid/chloroform + IAC AflaOchra HPLC^TM^	HPLC-FD	Kabak et al., 2009 [60]
Commercial pasta	DON	Static in vitro digestion model: Oral phase (KCl/KSCN/NaH_2_PO_4_/NaSO_4_/NaCl/NaHCO_3_/urea/a-amylase)Gastric phase (HCl/pepsin) Intestinal phase (NaHCO_3/_pancreatin/bile salts/H_2_O)	ACN:water (84:16; *v/v*)	LC-MS/MS	Raiola et al., 2012 [59]
Breakfast cereals Cookies Breads	ENA ENA1 ENB ENB1	Static in vitro digestion model: Oral phase (KCl/KSCN/NaH_2_PO_4_/NaSO_4_/NaCl/NaHCO_3_/urea/a-amylase) Gastric phase (HCl/pepsin) Intestinal phase (NaHCO_3/_pancreatin/bile salts/H_2_O)	Ethyl acetate	LC-DAD	Prosperini et al., 2013 [61]
Cereal-based food	PAT	Static in vitro digestion model: Oral phase (KCl/KH_2_PO_4_/NaHCO_3_/MgCl_2_(H_2_O)_6_/(NH_4_)_2_CO_3_) Gastric phase (KCl/KH_2_PO_4_/NaHCO_3_/NaCl/MgCl_2_(H_2_O)_6_/(NH_4_)_2_CO_3_/pepsin) Intestinal phase (KCl/KH_2_PO_4_/NaHCO_3_/NaCl/MgCl_2_(H_2_O)_6_/pancreatin/bile)	Ethyl acetate + sodium sulphate + sodium hydrogenocarbonate + SPE column	RP-HPLC-UV	Assunção et al., 2016 [46]
	OTA		MeOH:water (80:20) + IAC AflaOchra	RP-HPLC-FD	

AFB1—Aflatoxin B1; DON—Deoxynivalenol; ENA—Enniatin A; ENA1—Enniatin A1; ENB—Enniatin B; ENB1—Enniatin B1; OTA—Ochratoxin; PAT—Patulin; ACN—Acetonitrile; HPLC-FD —High-performance liquid chromatography coupled to fluorescence detector; HPLC-UV—High-performance liquid chromatography coupled to ultraviolet detector; IAC—Immunoaffinity columns; LC-DAD—Liquid chromatography coupled to diode array detector; LC-MS/MS—Liquid chromatography coupled to tandem mass spectrometry; RP—Reverse-phase; SPE—Solid-phase extraction.

## Data Availability

Not applicable.

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
