# Peer review of "Mycotoxins of Concern in Children and Infant Cereal Food at European Level: Incidence and Bioaccessibility"

_toxins, 2022, doi:10.3390/toxins14070488_

Round 1
Reviewer 1 Report
Please find reviewing comments as below:
In abstract:
Compared with UPLC-MS/MS and GC-MS, HPLC-FD has a lower price. If the cost of a single analysis and the sensitivity of the instrument are used to evaluate the method's preference, the cost of purchasing the instrument might be ignored.
In abstract:
This study did not consider that other foods for infants and young children other than cereals, such as fruit puree that may contain patulin, are not included in the main discussion items; and functional ingredients such as polyphenols and flavonoids in fruits inhibit the bioavailability of mycotoxins There is no discussion of power, and this part does not correspond to the content of the introduction (Line 35-36).
Linr 90:
You can follow other review papers to search for relevant literature using google scholar.
Line 110:
The incomplete metabolic development of infants and young children means that the bioavailability is lower or higher? The detoxification ability is stronger or weaker? None of the above is clearly explained, and it is inconsistent with the following description.
Line 112:
Why do you want to explore the "bioaccessibility" or "bioavailability" of infants and young children without searching for relevant epidemiological journals to prove that there are few cases of diseases caused by other infants and young children and the low incidence of diseases?
Table 2:
Average concentrations may be affected by outliers. It is recommended to describe this situation in the text or to add a field for median concentration. When comparing the positive rate, the LOD and LOQ of the detection methods of different literatures should be listed to avoid the low concentration of mycotoxin-containing samples being ignored, resulting in misjudgment of the positive rate.
Line 279-294:
The level of bioaccessibility may be related to whether the transport proteins in vivo has reached saturation, so it does not follow the dose-dependent effect.
Table 4:
There is no literature on in vitro testing of simultaneous consumption of multiple food matrices. It is recommended to re-search the literature to explain or clarify the effect of multiple food matrices on the bioavailability of mycotoxins.
Line 673:
It is mentioned that dynamic digestion models may be widely used, but so far there is no relevant literature review. How to confirm its applicability?
Reference:
Some cited journals are not abbreviated in ISO4 format, such as Journal of Human Nutrition and Dietetics = J Hum Nutr Diet. It is recommended that the full text content should be consistent in ISO4 format.
Others:
The proportion of infants and young children allergic to gluten in cereals is relatively high in Europe. It is recommended to discuss the positive rate of mycotoxins together with non-cereal foods for infants and young children.
Author Response
In abstract:
Compared with UPLC-MS/MS and GC-MS, HPLC-FD has a lower price. If the cost of a single analysis and the sensitivity of the instrument are used to evaluate the method's preference, the cost of purchasing the instrument might be ignored.
The sentence in the abstract was modified to: “LC-MS/MS is the method of choice for detection and quantification of mycotoxins due to its high sensibility and accuracy.”
In abstract:
This study did not consider that other foods for infants and young children other than cereals, such as fruit puree that may contain patulin, are not included in the main discussion items; and functional ingredients such as polyphenols and flavonoids in fruits inhibit the bioavailability of mycotoxins There is no discussion of power, and this part does not correspond to the content of the introduction (Line 35-36).
Our article focused on cereal-based food products intended for children. In order to highlight the importance of cereal in children nutrition we added the food groups of importance for a healthy and diversified nutrition.
Linr 90:
You can follow other review papers to search for relevant literature using google scholar.
Scholar google has been added as database in the materials and methods.
Line 110:
The incomplete metabolic development of infants and young children means that the bioavailability is lower or higher? The detoxification ability is stronger or weaker? None of the above is clearly explained, and it is inconsistent with the following description.
Due to their underdeveloped metabolism, children are more susceptible to exposure to mycotoxins and have weaker detoxification capacity. Therefore, children are more prone to acute mycotoxicosis. The long-term effects, chronic mycotoxicosis, are usually seen in adults, and not children, due to the low levels of mycotoxins in foods and changes in diet as the children grow.
A clarification was added as requested.
Line 112:
Why do you want to explore the "bioaccessibility" or "bioavailability" of infants and young children without searching for relevant epidemiological journals to prove that there are few cases of diseases caused by other infants and young children and the low incidence of diseases?
The bioavailability and bioaccessibility of mycotoxins are an important factor to evaluate their health effects in both adults and children. As children’s digestive systems are underdeveloped and they may be more susceptible to the acute effects of mycotoxins, it is important to develop methods targeting infants digestibility in order to evaluate the real secondary effects of mycotoxins exposure in children.
Table 2:
Average concentrations may be affected by outliers. It is recommended to describe this situation in the text or to add a field for median concentration. When comparing the positive rate, the LOD and LOQ of the detection methods of different literatures should be listed to avoid the low concentration of mycotoxin-containing samples being ignored, resulting in misjudgment of the positive rate.
The LOD and LOQ of each study are added in the Supplemental Table 1.
Line 279-294:
The level of bioaccessibility may be related to whether the transport proteins in vivo has reached saturation, so it does not follow the dose-dependent effect.
The cited papers did not refer anything about saturation of the in vivo proteins transport.
Table 4:
There is no literature on in vitro testing of simultaneous consumption of multiple food matrices. It is recommended to re-search the literature to explain or clarify the effect of multiple food matrices on the bioavailability of mycotoxins.
The articles in table 4 are the only we found on bioavailability of cereal-based children food products.
Line 673:
It is mentioned that dynamic digestion models may be widely used, but so far there is no relevant literature review. How to confirm its applicability?
Dynamic digestion models offer a more realistic approach to the digestion process, however they are more complicated, time-consuming and have a higher cost. With advancements in automation process and continuous research of the effects of mycotoxins, the authors believe that dynamic digestion models may be easier to replicate and be more cost-efficient, leading to a wider use of these methods.
Reference:
Some cited journals are not abbreviated in ISO4 format, such as Journal of Human Nutrition and Dietetics = J Hum Nutr Diet. It is recommended that the full text content should be consistent in ISO4 format.
The references were reviewed as suggested.
Others:
The proportion of infants and young children allergic to gluten in cereals is relatively high in Europe. It is recommended to discuss the positive rate of mycotoxins together with non-cereal foods for infants and young children.
The authors thank you kindly for the suggestion of improvement of our paper; we felt that a targeted review towards cereal-based food products was important to raise awareness to the subject of mycotoxins in a food product that is so widely consumed.

Reviewer 2 Report
The reviewed manuscript presented for its publication in Toxins deserves to be accepted as it gives a general overview of the exposure to mycotoxins by children... However it is too large and some sections must be avoided, as it is dealing with bioaccessibility and incidence and having sections about the determinations, type of equipments used for the determination is out of place and there are already reviews of chromatography dealing with it. In this sense, the sections 7.1.1, 7.1.2, and 7.1.3 must be deleted.
Please, consider that this is a very long review and including co-lateral sections is resulting more confusing than helpful.
Author Response
Dear reviewer, the authors thank you kindly for your suggestion of improvement of our
manuscript. We feel that the sections on determination and instrumentation are an
important complement of our paper since we debate the incidence as well as the
bioaccessibility of mycotoxins in cereal-based products intended for infants and
children, also, the other reviewers have also agreed and liked the structure of our paper.

Reviewer 3 Report
Interesting and very well written article. Some attention should be devoted to editorial issues. I don't really understand why table 2 was split, in the middle of the page. Maybe it is better to leave the table continuous.
From the suggestions the authors may consider, it seems to me that a figure or table would be useful to show how mycotoxins act on the body. A little more attention to the mechanisms and target organs against which mycotoxins are harmful would be advisable.
Minor comments:
line 52-53 and whole manuscript: nomenclature names should be written in italics. in these cases the name of the genera of fungus in italics, e.g. Fusarium, and the term spp. without italics.
Table 1: proposes in the table, in addition to abbreviations, the full names of mycotoxins to be given
line 667: this information shows that there should be a table in the supplement. This needs to be verified
The whole article needs to be re-read, I found some errors with commas and double spaces. This has to be eliminated at this stage
Author Response
Reviewer 2
Interesting and very well written article. Some attention should be devoted to editorial issues. I don't really understand why table 2 was split, in the middle of the page. Maybe it is better to leave the table continuous.
Table was altered as suggested.
From the suggestions the authors may consider, it seems to me that a figure or table would be useful to show how mycotoxins act on the body. A little more attention to the mechanisms and target organs against which mycotoxins are harmful would be advisable.
A new figure was added to the manuscript showing how mycotoxins act on the body.
Minor comments:
line 52-53 and whole manuscript: nomenclature names should be written in italics. in these cases the name of the genera of fungus in italics, e.g. Fusarium, and the term spp. without italics.
The document was reviewed as requested.
Table 1: proposes in the table, in addition to abbreviations, the full names of mycotoxins to be given
The changes were added as requested.
line 667: this information shows that there should be a table in the supplement. This needs to be verified
The supplemental table was added.
The whole article needs to be re-read, I found some errors with commas and double spaces. This has to be eliminated at this stage
The article was reviewed as requested.
Round 2
Reviewer 2 Report
The main point suggested inthis good review was to suppress the section comprised in section 7.1 where all the determination is explained.
It is necessary to delete all this section as there are already other reviews containing this and it is not the scoupus of the review. it s intend to enlarge the content while the review is complete enough with the other sections. Again: DELETE SECTION 7.1
Another issue detected is the prsence of mycotoxins in cereals. Please, reduce section 5.
With this two points of redcution and redunctant information, the review could be accepted; otherwise any of the reuirements and/or suggestions are considered...
In summary, the manuscript passes now to Reconsider after major revision.
Author Response
The main point suggested in this good review was to suppress the section comprised in section 7.1 where all the determination is explained.
It is necessary to delete all this section as there are already other reviews containing this and it is not the scoupus of the review. it s intend to enlarge the content while the review is complete enough with the other sections. Again: DELETE SECTION 7.1
The section 7.1 was moved to the supplementary material. Suggestion accepted.
Another issue detected is the prsence of mycotoxins in cereals. Please, reduce section 5
The section makes an overview of mycotoxin contamination in cereal products for infants, so we prefer not to move it to the supplementary, because in this
manner the reader can immediately become aware of the contamination differences. No modification has been made to the text.